# Effects of Y_2_O_3_ and LiAl_5_O_8_ on the Microstructure and Optical Properties of Reactively Sintered AlON Based Transparent Ceramics

**DOI:** 10.3390/ma15228036

**Published:** 2022-11-14

**Authors:** Guojian Yang, Peng Sun, Yuezhong Wang, Zitao Shi, Qingwei Yan, Shasha Li, Guoyong Yang, Ke Yang, Shijie Dun, Peng Shang, Lifen Deng, He Li, Nan Jiang

**Affiliations:** 1Key Laboratory of Marine Materials and Related Technologies, Ningbo Institute of Materials Technology and Engineering, Chinese Academy of Sciences, Ningbo 315201, China; 2Center of Materials Science and Optoelectronics Engineering, University of Chinese Academy of Sciences, Beijing 100049, China; 3Tianjin Key Laboratory of Optical Thin Films, Tianjin Jinhang Technical Physics Institute, Tianjin 300192, China

**Keywords:** transparent ceramic, AlON, sintering aid, reaction sintering, residual pore

## Abstract

Sintering aid was very crucial to influence the microstructure and thus the optical property of the sintered optical ceramics. The purpose of this work was to explain the difference between the sintering aids of Li^+^ and Y^3+^ on Al_23_O_27_N_5_ (AlON) ceramic via reaction sintering method. The effects of LiAl_5_O_8_ (LA) and Y_2_O_3_ on the sintering of Al_2_O_3_–AlN system were carefully compared, in terms of X-ray diffraction (XRD), microstructure, density, X-ray photoelectron spectroscopy (XPS) and optical transmittance. According to the XPS and XRD lattice analysis, the chemical structure of the materials was not obviously affected by different dopants. We firstly reported that, there was obvious volume expansion in the Y^3+^ dopped AlON ceramics, which was responsible for the low transparency of the ceramics. Obvious enhancements were achieved using Li^+^ aids from the results that Li: AlONs showing a higher transparency and less optical defects. A higher LA content (20 wt%) was effective to remove pores and thus obtain a higher transmittance (~86.8% at ~3.5 μm). Thus, pores were the main contributions to the property difference between the dopant samples. The importance of sintering aids should be carefully realized for the reaction sintering fabrication of AlON based ceramics towards high transparency.

## 1. Introduction

Highly transparent spinel-type Al_23_O_27_N_5_ (AlON) ceramic is an attractive polycrystalline material that exhibits many outstanding advantages (such as low cost, excellent optical and mechanical properties [1,2]). It was firstly synthesized by Yamaguchi and Yanajida in 1959 [3]. AlON was a solid-solution with the composition that centered at Al_23_O_27_N_5_ (or 9Al_2_O_3_·5AlN) [4]. It showed processing flexibility in fabricating large size and complex shape and exhibited wide range of optical transmittance. Therefore, AlON transparent ceramics were capable for ideal infrared windows, domes and transparent armors [5,6].

The AlONs showed thermodynamically instability below ~1640 °C [7], and thus a higher synthesis temperature should be required. After the fabrication of the first translucent AlON ceramic in 1979 [8], two methods have been developed to prepare AlON ceramics with high transparency over the decades. The first was the pressureless sintering method using AlON powder as raw materials. The other was reaction sintering using AlN and Al_2_O_3_ powders. To approaching AlON’s theoretical density, sintering aids (such as Y_2_O_3_, La_2_O_3_ and MgO) have been widely used [9,10,11,12], which played significant roles in reducing pores and promoting densification. Very small amounts of the dopants could be used due to the extremely limited solubility in AlON. Consequently, they could enrich or precipitate on grain boundaries and reduce transmittance obviously. The solubility limits of La^3+^ and Y^3+^ were reported to be only 498 ± 82 and 1775 ± 128 ppm in AlON by sintering at the temperature as high as 1870 °C by Miller and Kaplan [13]. Using the presureless sintering method, Wang et al. [14] reported using Y_2_O_3_ and La_2_O_3_ as a composite sintering additive. The obtained AlON ceramics showed high transparency (~80.3%) at 400 nm wavelength by using 0.12 wt% Y_2_O_3_ and 0.09 wt% La_2_O_3_. Y_2_O_3_ and MgO co-doped AlON with high transmittance (~86.1%) was also successfully fabricated by Jiang et al. [15] via pressureless sintering (1800 °C, 6 h) followed by hot isostatic pressing (HIP) at 1825 °C for 3 h. High transmittance was mainly benefited from thorough elimination of residual pores and secondary phases. However, the using of Y_2_O_3_ or La_2_O_3_ dopants was not so successful for the reaction sintering method. Study [16] indicated that Y_2_O_3_ below 0.6 wt% could extremely promote the sintering reaction. However, excess Y_2_O_3_ would react with Al_2_O_3_ to form Y_3_Al_5_O_12_ (YAG) phase which could limit the grain growth of AlON ceramics. Therefore, major obstacles for the elimination of optical defects, such as pores and secondary phases (due to low solid solution in AlON) during reaction sintering, still remained to be overcome. Furthermore, a high temperature (≥ 1850 °C) was still inevitable using the reaction sintering method with Y^3+^ dopant. HIP was normally used as a post-processing method to further reduce the pores [17,18].

To obtain high transparency and decrease the sintering temperature, others additives with high solid solution limits in AlON were adopted, such as MgO and LiAl_5_O_8_ (LA). It was suggested that AlON could be stabilized using MgO at a lower temperature [19]. As early as 1994, first transparent Mg: AlON ceramic was fabricated by Granon et al. [20] via reaction sintering and post-HIP. High infrared (IR) transmittance (~80%) was obtained but the visible band was not high enough (~65%). Higher transparency (~84%) across the whole band of Mg: AlON was not reported until 2014 by Liu et al. [21] using the pressureless sintering method. The reported strength (~280 MPa) and hardness (~13.5 GPa) of Mg: AlON are slightly lower than those of AlON (~310 MPa, 15 GPa) due to ~8% MgO doping [22,23]. Previously, Li: AlON with a higher hardness (~15–17 GPa) had been produced by Clay et al. [24] by reaction sintering using LiAl_5_O_8_ (LA, also called as zeta alumina), Al_2_O_3_ and AlN. However, the optical transmittance (~ 65%) was not high enough for practical application. Generally, transmittance loss could be attributed to light scattering centers, such as pores, secondary phases and impurities [25,26]. Highly transparent Li: AlON without obvious scattering sources was reported in 2018 by Zhang et al. [27]. They prepared a Li: AlON ceramic with the maximum transmittance of ~85.5% in a broadband from visible to mid-infrared band by a two-step method (reaction sintering and post-HIP). The samples also showed a higher flexural strength (~310 MPa). In our previous work [28], Li: AlON with higher transmittance (~86.8%) and flexural strength (~332 MPa) was fabricated by reaction sintering of LA–Al_2_O_3_–AlN composites. The work compared the sintering of LA doped and aid-free AlON ceramics. It was suggested that the LA could depress the volume expansion comparing that of dopant-free system.

Numerous attentions were attracted to the fabrication of AlON ceramics towards high transparency. However, the obtaining of Y^3+^ doped AlON with high transparency is still a hard task by the reaction sintering method. Related mechanism on the microstructural densification has not been well investigated. Consequently, a further study on the widely used aids (Y^3+^ and Li^+^) is very essential for AlON ceramic fabrication. Presently, it is of a primary goal to identify the difference by which the microstructure and transparency are influenced in this type of materials. Therefore, a comprehensive comparison between Y_2_O_3_ and LA dopants for AlON ceramic fabrication by reaction sintering was carried out. The density, phase, microstructure and optical transmittance of obtained Y: AlON and Li: AlON were carefully studied.

## 2. Experiments

### 2.1. Composition Design and Ceramic Fabrication

Y: AlON and Li: AlON ceramic samples were prepared by reaction sintering under nitrogen atmosphere. The raw powders and the producers: Al_2_O_3_ (Sumitomo Chemical Co., Ltd., Tokyo, Japan), Y_2_O_3_ (Alfa Aesar, Haverhill, MA, USA), AlN and LiAl_5_O_8_ (Ultrapure Applied Materials Co., Ltd., Chengdu, China). The mixed powders were consisted of AlN (10 wt%), Al_2_O_3_ (varying composition range of 70–90 wt%), Y_2_O_3_ (0.5 wt% and 1 wt%, respectively) and LiAl_5_O_8_ powders (10 wt% and 20 wt%, respectively). The average particle sizes of AlN, Al_2_O_3_, Y_2_O_3_ and LiAl_5_O_8_ raw powder were approximately 1 μm, 150 nm, 100 nm and 200 nm, respectively. These mixed powders of AlN–Al_2_O_3_–Y_2_O_3_ were divided into two groups denoted as the corresponding A and B samples. For comparison, the AlN–Al_2_O_3_–LiAl_5_O_8_ compositions of C and D were designed to ensure the same amounts of Li_2_O comparing Y_2_O_3_ according to Equation (1) [24,29]. The compositions of A–D and the theoretical densities (g/cm^3^) were shown in Table 1.
2LiAl_5_O_8_ = Li_2_O + 5Al_2_O_3_(1)

The powders were mixed by planetary-ball milling method using absolute ethanol as dispersion medium. One vertical planetary-ball milling grinder (Model YXQM, Mitr Instrument Equipment Co., Ltd., Changsha, China) with a constant ratio of 2:1 of rotating rated of the jar and sun disk was used in this study. To control the degree of contamination, high quality alumina balls (3.7 g/cm^3^, 99.99%, Φ10 mm, Jingdezhen Betterwear New Materials Co., Ltd., Jingdezhen, China) and nylon jar were used. The ratio of ball to powder was 8:1. The milling time and rate were 24 h and 200 rpm, respectively. After milling, the slurry was dried in vacuum at 50 °C for 2 h in a rotary evaporator to evaporate the ethanol, followed by sieved with a 100-mesh sieve. After sieving, they were uniaxially dry pressed (6 MPa for 5 min) and then cold isostatically pressed (CIP, 220 MPa for 10 min) into Φ20–70 mm × 4–10 mm pellets. The obtained green bodies were then calcined at 650 °C for 20 h in air in a muffle furnace to eliminate organic residues. Then, the samples were sintered at 400–1800 °C (with the heating rate of 10 °C/min) and kept for 2 h under a N_2_ atmosphere in a graphite heating furnace. The cooling rates were approximately 10 °C/min (free cooling below 600 °C). To avoid potential pollution, all the samples were buried with boron nitride (BN) powder (99.99%, ~100 nm) in graphite crucibles. To obtain optical transparency, preferred samples were further treated by HIP (1880 °C for 4 h) under the pressure of 185 MPa Ar atmosphere. All the samples were mirror polished on both sides for the experimental characterizations.

### 2.2. Materials Characterization

The phases of ceramic samples were identified by X-ray diffraction (XRD, Advance D8, Brucker, Bremen, Germany) using filtered CuKα radiation. To avoid hydrolysis, the densities of the samples were measured by the geometrical method. The microstructure was tested by a scanning electron microscopy (SEM, 8230, Hitachi, Tokyo, Japan). The chemical states of obtained samples with different doping were identified by X-ray photoelectron spectroscopy (XPS, Axis Ultra, Kratos, Manchester, UK). The in-line transmittance spectra were measured by ultraviolet-visible spectroscopy from 200 to 2500 nm (UV–VIS, Model Lambda-900, PerkinElmer, Waltham, MA, USA) and Fourier transform infrared spectroscopy from 2500 nm to 6500 nm (FT–IR, Model Nexus, Thermo Nicolet Corporation, Madison, WI, USA).

## 3. Results and Discussion

### 3.1. Sintering and Densification

Figure 1 showed the densities variation and appearance of the samples sintered at different temperature. As shown in Figure 1a, there was no obvious density change in all samples until 1200 °C. The sintering and densification could be in the initial stage. Above 1200 °C, a significant increase of the densities was seen in all compositions with the increasement of temperature. For the Y_2_O_3_-doped systems, there was no obvious difference of the density curves between sample A and sample B. For both samples, the densities increased with temperature and reached the highest values (nearly 3.85 g/cm^3^) at 1550 °C. At higher temperatures, they gradually lowered to ~3.62 g/cm^3^ (relative density ~98.6% of AlON) and then remained constant. The decreasing of densities at a high temperature was indicative of volume expansion, since no obvious mass changes were observed in the samples. For the LA-doped compositions C and D, there was an obvious difference between them above 1200 °C. At 1200–1500 °C, densification was benefited from a lower content of LA, as shown in Figure 1a. However, the situation was reversed at a higher temperature of 1550–1700 °C where volume expansion appeared for the 10% LA–AlN–Al_2_O_3_ system. It was suggested that the volume expansion during sintering could not be avoided by a low content of LA. The density of sample D continuously increased at 1200–1700 °C, which indicated that the volume expansion of this system was absolutely inhibited by high content LA-doping. Notably, all curves were nearly flat at 1750 °C, reaching ~98.6% of the theoretical density, which indicated that both reaction and main densification were nearly completed. Depressed densification or volume expansion could be attributed to the chemical reaction between AlN and Al_2_O_3_ [30]. The role of LA could be accounted by the fact that it could transform into a face-centered cubic structure at above 1290 °C with a similar lattice parameter (a = 7.920 Å) to that of AlON [31]. The addition of LA could not enhance sintering but reduce the number of scattering resources by pore coalescence [27]. In Figure 1b, the dimensional change was consistent with the trend of the curve in Figure 1a. Evidently, significant color differences could be found between these four systems. The samples sintered at 1550–1650 °C were all white and seemed almost the same. As the temperature rose, samples A and B apparently became dark at 1700 °C and darker at 1750 °C but slightly lighter at 1800 °C. In contrast, no obvious color changes were observed in C and D. It indicated that the addition of LA could promote the elimination of discoloration due to carbon contamination or lattice defects during sintering under a high temperature [32,33].

### 3.2. Phase and Microstructural Evolution

To better understanding the densification behaviors, phase evolutions of the samples were investigated at the temperature in 1550–1800 °C. As shown in Figure 2a–d, AlON had not been formed in all samples at 1550 °C. It indicated that the densification of all samples before 1550 °C in Figure 1a could be mainly attributed to the shrinkage of the raw powders (early stage of sintering) [34]. As temperature increased to 1600 °C, the Y-doped and 10 wt% LA-doped systems (Figure 2a–c) were still mainly composed by AlN and Al_2_O_3_. However, a slight decrease of the intensity of AlN and Al_2_O_3_ phases were seen (Figure 2a–c), indicating of the formation of trace of AlON that were escaped from the detecting limits of the equipment. For the observation, the subsequent decreases of densities from 1550 °C to 1600 °C in Figure 1a could be attributed to the formation of AlON phases. Meanwhile, volume expansion occurred due to the solid-state reaction between AlN and Al_2_O_3_. The addition of 10 wt% LA was no enough to depress the volume expansion. The situation was improved for 20 wt% LA addition, as was shown in Figure 2d, where Li: AlON was obviously formed at 1600 °C. The formation temperature of AlON phase was obviously affected by the LA content and was decreased by 50 °C using 20 wt% LA, below which the effect was not obvious (as for 10 wt% LA). As the temperature increased, AlN and Al_2_O_3_ disappeared gradually, and AlONs continuously formed both for Y_2_O_3_–AlN–Al_2_O_3_ system and LA–AlN–Al_2_O_3_ systems. As the temperature increased from 1650 °C to 1700 °C, AlN phases almost disappeared. It could be suggested that nitrogen-rich AlON phases firstly formed and then reacted with residual Al_2_O_3_. The situation was different from previous reports [35]. The difference could be attributed to higher Al_2_O_3_ contents of the raw mixtures in our work. For the Y-doped and 10 wt% LA-doped systems (Figure 2a–c), AlONs formed and then became single phases at 1750 °C. For 20 wt% LA-doped system, single Li: AlON phase was obtained at 1700 °C (Figure 2d), which was 50 °C lower than that of the previous three compositions. XRD patterns of single AlON phases of four samples exhibited good similarity with previous reports on Y: AlON [14,16] and Li:AlON [27]. It indicated that the addition of LA at a higher content (20 wt%) could effectively promote the chemical reaction between AlN and Al_2_O_3_. Lower content had no obvious promotion effect.

Further insights into sintering and densification of the systems were studied by observing microstructural evolution. For the Y_2_O_3_ doped samples, there was no obvious difference between the two compositions (Figure 3a,b). Very few pores were detected at 1500–1650 °C, which were accounted by the high densities (Figure 1a). Grain coarsening was also observed at this stage. It seemed that the densification was promoted by grain coarsening for the Y_2_O_3_ doped systems in this sintering stage. However, the promotion effects were prohibited due to microstructural large pores formation at a higher temperature (1700 °C). There were obvious differences between the Y_2_O_3_ doped and LA doped samples. Firstly, the grain sizes of samples C and D were significantly smaller than that of A and B at 1500–1650 °C. Anticipatedly, more pores (low densities) were observed in these samples (Figure 3c,d). No obvious difference was observed between the 10 wt% LA and 20 wt% LA doped samples at 1500–1600 °C. In contrast, a slight difference was presented at 1650 °C. The sample of 20 wt% LA doped sample was less porous. It was therefore suggested that the migration of grain boundaries was promoted by the addition of Y_2_O_3_, or depressed by LA doping at this stage. Secondly, at a higher temperature of 1700 °C, obvious grain coarsening was also observed, with no larger residual pores. In contrast to the 10 wt% LA doped sample, the 20 wt% LA doped ceramic showed a relative coarser microstructure at 1700 °C. For reactively sintered systems, the microstructure and thus the optical properties were very susceptible to temperature and dopants [36,37,38]. In combination with the results (Figure 2 and Figure 3), it was concluded that LA could prevent the sintering at a lower temperature (below 1650 °C). However, the sintering process was obviously facilitated at a higher temperature [26]. The role of Y_2_O_3_ was conversed comparing that of LA.

### 3.3. Optical Properties and Material Structure

Transparent ceramics were fabricated by reaction sintering at 1700 °C for 24 h followed by a HIP treatment (1880 °C for 4 h). Figure 4a–d showed the appearances of obtained samples with the sizes of ~Φ18 mm × 2 mm. As seen, both the 0.5 wt% and 1.0 wt% Y_2_O_3_-doped ceramics were translucent with depressed transparency where the words behind them were dark in color. In contrast, the transparency of LA-doped samples showed an obvious improvement as shown in Figure 4c,d. The words behind the samples were clearly seen. Enhanced transparency was resulted from 10 wt% to 20 wt% LA dopants. Figure 4e showed the transmittance of all samples. The highest transparency was shown in the 20 wt% LA dopped sample with ~87% transmittance in a wide band. With 10% LA addition, the transmittance from the visible to infrared bands presented obvious withdraw (with the maximum of ~82% at ~3.7 μm). For the Y_2_O_3_ doped AlONs, the transmittances were obviously depressed, especially in the visible bands. The highest transmittances were ~73% and ~61% for the 0.5% Y_2_O_3_ and 1% Y_2_O_3_ doped systems, respectively. Compared with the additive-free sample [18,28], Y_2_O_3_ doping was effective to enhance the transparency, however, the effect was limited to some degree. Notably, all samples reached their maximum transmittances at ~3.7 μm and the cutoff bands ~6 μm due to their similar intrinsic performance. It indicated that the depressed transmittances were mainly caused by the presence of micro-defects, such as residual pores and secondary phases. Figure 4f showed that all samples were single phases without obvious secondary phases. For the consideration, the main scattering sources were therefore suggested to be the residual pores. Further insights into the chemical structure of the ceramics were investigated by XRD lattice measurement and the XPS spectra. The XRD parameters of four obtained samples in Figure 4a–d have been calculated, as shown in Table 2. The lattice parameters and volume were approximately 7.93–7.94 Å and 499–500 Å^3^, respectively. The network parameters showed no obvious difference between the compositions. According to previous publication, the parameters of AlONs were mainly dependent on N contents [39,40]. The lower AlN content, the lower lattice parameter and volume. Therefore, a slight difference between the parameters was perhaps due to minor composition difference. As shown in Figure 5a, there was also no obvious difference between the XPS patterns. The bonding energy of Al 2p were 74.32, 74.28, 74.28 and 74.38 eV (Figure 5b), respectively. According to the deep investigation [41], the Al 2p and N 1s XPS showed that the AlON film was composed of Al−N, Al−O, and N−Al−O bonds. These bonding energy values were also very close to the literature reports [41,42,43]. It seemed that the chemical structure of the materials was not obviously changed by these dopants. Additionally, the shorter cutoff wavelength was near 0.23 μm for all samples, therefore the bandgaps (*E*_g_) of the materials were ~5.2 eV according to the equation *E*_g_ = *hc*/*λ*_min_.

Obvious differences of the transmittances in these samples could be attributed to the vary microstructure caused by different doping conditions. Further study of the microstructure supported the conclusion that residual pores were responsible for the depressed transmittances. As shown in Figure 6a,b, plenty of pores were located at the grain boundaries in samples A and B. The residual pores could defeat the transmittance as scattering centers. In contrast, there were much fewer pores in sample C and almost no obvious pores in sample D (Figure 6c,d). Furthermore, the densities of these sintered samples after HIP showed a slight increase from 3.69, 3.69, 3.68 and 3.68 g/cm^3^ to 3.70, 3.70, 3.70 and 3.69 g/cm^3^, respectively. Lower porosity was hence supported by a higher related density (~100%) of LA doped systems. Different dopants showed different degrees of influences on the microstructural evolution and thus the optical transmittance [44,45,46]. For the present investigation, 20%-LA doping exhibited the best effect to promoting densification and eliminating the residual pores. In combination with the above XRD, SEM and XPS analysis, we could obtain the conclusion that: (a) the dopants effected only the microstructural defect (pores) and thus optical transmission; and (b) the dopants did not obviously change the phase, chemical binding and lattice parameters of the obtained ceramics.

According to the above results, a larger sized (Φ55 mm × 6 mm) Li: AlON ceramic with high transparency was fabricated. The sintering was carried out at 1750 °C for 24 h followed by a HIP treatment (1880 °C for 4 h). After optical polishing on both sides, the obtained ceramic was water-clear and the leaves behind it were clearly visible (Figure 7a). The transmittance reached ~86.8% at ~3.5–3.7 μm and >80% at a wide band range (Figure 7b). In addition, the absorption near 3.25 μm was considered to be caused by -OH formed by the adsorption of oxygen and water molecules in air on sample surface [47]. Moreover, we fabricated AlON based ceramics with different transparency, and related mechanism on the effects of dopants were systematically investigated. Li: AlON with high transparency was obtained. Precise measurement of optical constants (n, k, Eg, etc.) could be anticipated in our next work for more insights into this material. Related methods have been successfully conducted with thin film materials [48,49].

## 4. Conclusions

(1)For the Y_2_O_3_ doped systems, pores could be effectively removed at the early sintering stage (1500–1650 °C); the densification was promoted by grain coarse at this stage. However, the densification was prohibited at a higher temperature accomplished by volume expansion, and large residual pores resulted from chemical reaction;(2)For the LA doped systems, pores could not be effectively removed at the early sintering stage (1500–1650 °C); the densification was prohibited by LA at this stage. The densification was promoted at a higher temperature. Composition with a higher LA content (20 wt%) showed a continuous densification during the sintering;(3)The chemical structure of the materials was not obviously affected by different dopants. Pores were the main contribution for the difference between the samples;(4)A larger size of Li: AlON transparent ceramic via reaction sintering method was obtained using 20 wt% LA doping. The ceramic has a pore-free microstructure and excellent optical transmittance (~86.8% at ~3.5 μm).

## Figures and Tables

**Figure 1 materials-15-08036-f001:**
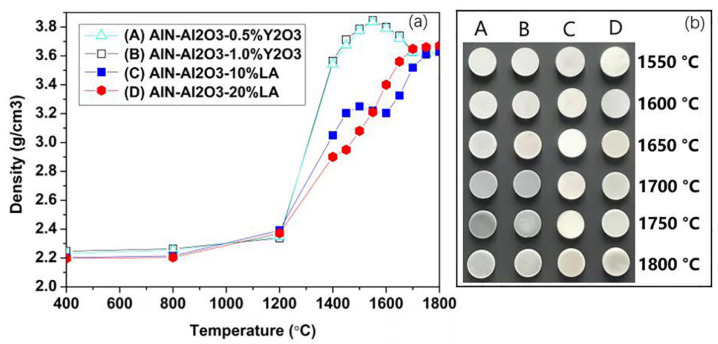
The densities (**a**) and photos (**b**) of the samples that sintered under different temperatures.

**Figure 2 materials-15-08036-f002:**
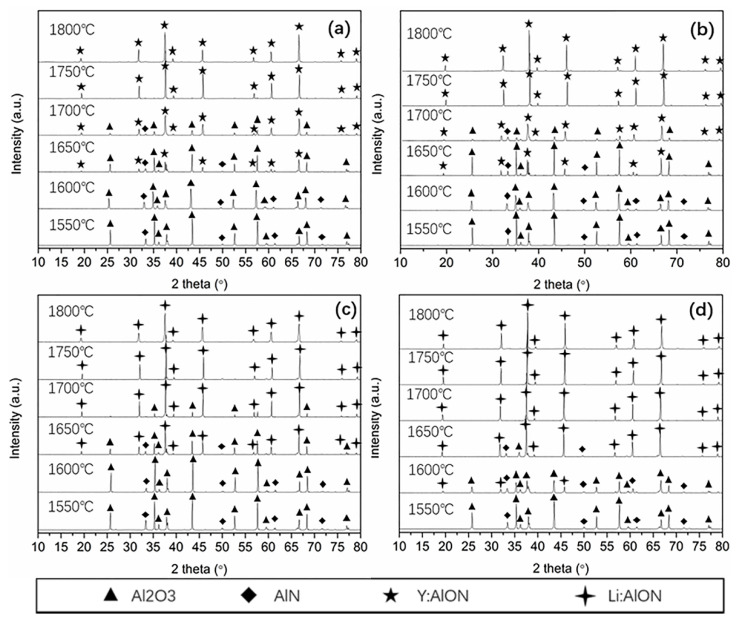
XRD of the ceramics sintered under different temperatures: (**a**) AlN–Al_2_O_3_–0.5% Y_2_O_3_; (**b**) AlN–Al_2_O_3_–1.0% Y_2_O_3_; (**c**) AlN–Al_2_O_3_–10% LA and (**d**) AlN–Al_2_O_3_–20% LA.

**Figure 3 materials-15-08036-f003:**
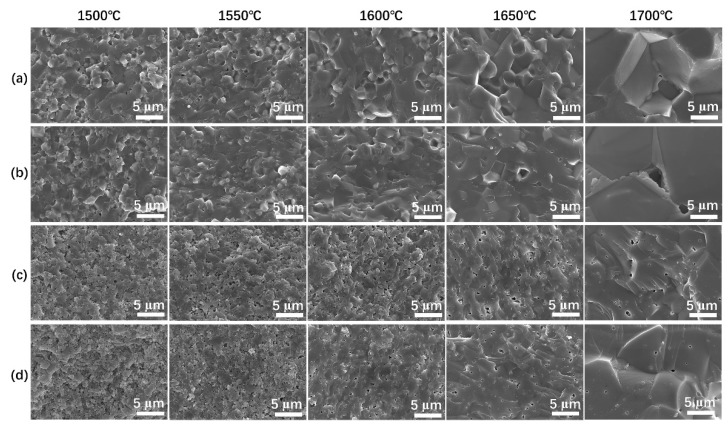
SEM of the fracture surfaces of the ceramics sintered under different temperatures: (**a**) AlN–Al_2_O_3_–0.5% Y_2_O_3_; (**b**) AlN–Al_2_O_3_–1.0% Y_2_O_3_; (**c**) AlN–Al_2_O_3_–10% LiAl_5_O_8_ and (**d**) AlN–Al_2_O_3_–20% LiAl_5_O_8_.

**Figure 4 materials-15-08036-f004:**
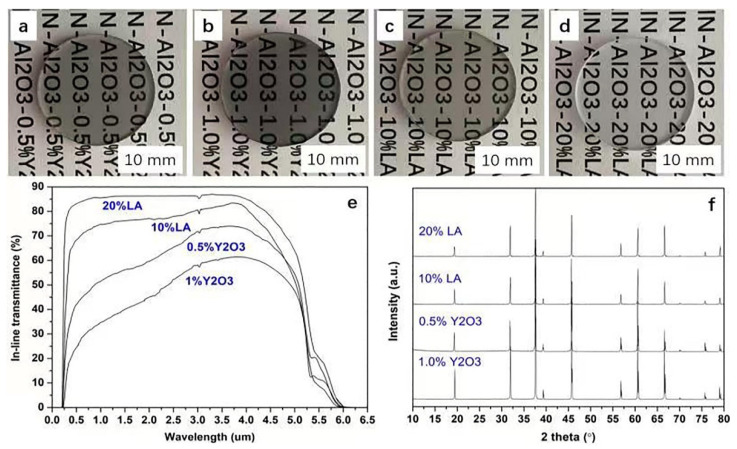
(**a**–**d**) Photos of the ceramics from AlN–Al_2_O_3_–0.5% Y_2_O_3_, AlN–Al_2_O_3_–1.0% Y_2_O_3_, AlN–Al_2_O_3_–10% LA and AlN–Al_2_O_3_–20% LA, respectively; (**e**) the corresponding transmittance and (**f**) XRD.

**Figure 5 materials-15-08036-f005:**
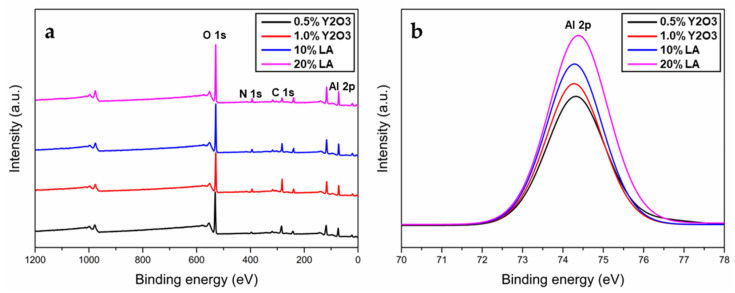
(**a**) XPS spectra of the samples with different dopants and (**b**) high-resolution XPS spectra of Al 2p.

**Figure 6 materials-15-08036-f006:**
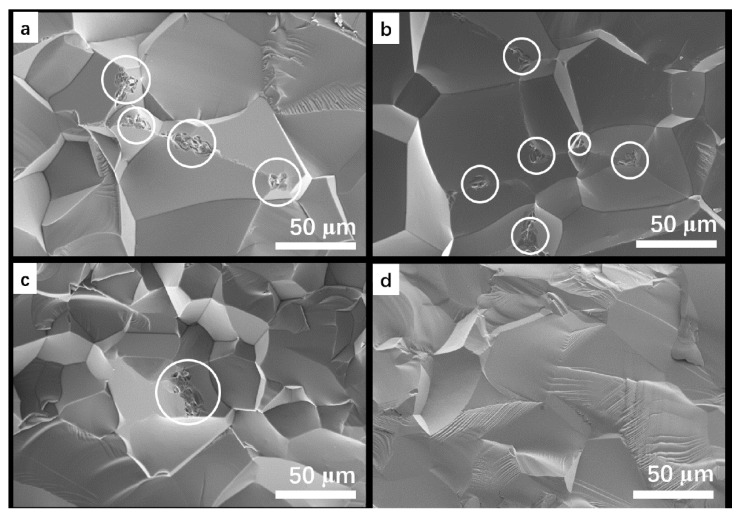
SEM of the fracture surfaces of the obtained ceramics: (**a**) AlN–Al_2_O_3_–0.5% Y_2_O_3_; (**b**) AlN–Al_2_O_3_–1.0% Y_2_O_3_; (**c**) AlN–Al_2_O_3_–10% LA and (**d**) AlN–Al_2_O_3_–20% LA. Residual pores were shown in the circles.

**Figure 7 materials-15-08036-f007:**
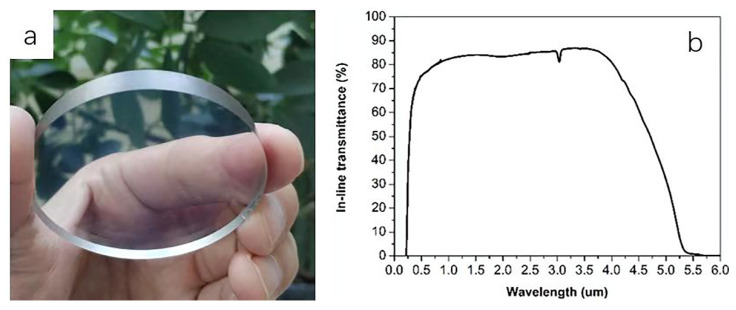
(**a**) Li: AlON transparent ceramic (Φ55 mm × 6 mm) and (**b**) the optical transmittance.

**Table 1 materials-15-08036-t001:** Composition design for the sintering experiments.

Materials	A	B	C	D
AlN	10%	10%	10%	10%
Al_2_O_3_	90%	90%	80%	70%
Y_2_O_3_	0.5%	1%	-	-
LiAl_5_O_8_	-	-	10%	20%
Theoretical density	3.71	3.71	3.70	3.69

**Table 2 materials-15-08036-t002:** Crystal lattice parameters of the AlON based ceramic samples with different dopants.

Samples	lattice Parameters (Å)	Volume (Å^3^)
0.5% Y_2_O_3_	7.9369	499.9801
1% Y _2_O_3_	7.9378	500.1502
10% LA	7.9330	499.2434
20% LA	7.9297	498.6206
AlONs [39]	7.9380–7.9550	500.1880–503.4085
AlON [40]	7.9380	500.1880

## Data Availability

The data that support the findings of this study are available from the corresponding author upon reasonable request.

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
