# Peer review of "Effects of Y2O3 and LiAl5O8 on the Microstructure and Optical Properties of Reactively Sintered AlON Based Transparent Ceramics"

_materials, 2022, doi:10.3390/ma15228036_

Round 1

Reviewer 1 Report

In this work, the authors investigate the effects of sintering aids on the microstructure and optical properties of reactively sintered AlON based transparent ceramics. The work is very interesting and with a high impact on literature. The manuscript can be accepted for the publication in Materials after addressing the following major comments.

1. The title i not very suggestive. Please reformulate.

2. Abstract: Describe more relevant results in the abstract. Mention the purpose for which this study was conducted.

3. The lengthy sentences may be split in to smaller sentence without change of its meaning.

4. Also, suggested to include the recent references in the introduction part.

5. The results and discussions part should be compared with the literature data. To redo the part of results and discussions by a systematic presentation of the results by which the readers of the articles manage to follow the article more easily.

6. Compare XRD results with other articles.

7. In discussions about structural analysis, XPS measurements are missing. Let these measurements be added to have a substrate for the correlation of the discussions.

8. Calculate the XRD parameters (crystallite size, network parameter, volume, etc.) and enter in a table. It can be discussed how the network parameters increase for the majority fax which becomes unique at higher temperatures

9. SEM figure quality is poor throughout. To improve the quality of the figures. Enlarge the characters in the figures.

10.Conclusions should be short with important observations.

11. References are not written according to the guide (not all authors are listed in all references, they are not marked with initials, the titles of the articles and magazines are not mentioned in many places, the volumes or pages are missing).

Author Response

Responses to reviewer #1

Comment 1: the title is not very suggestive. Please reformulate.

Response:

The title has been changed in the revised manuscript to “Effects of Y2O3 and LiAl5O8 on the microstructure and optical properties of reactively sintered AlON based transparent ceramics”.

Comment 2: Abstract: describe more relevant results in the abstract. Mention the purpose for which this study was conducted.

Response:

Related revisions have been made in the abstract. As shown next.

  • Line 16–20. “The effects of LiAl5O8 (LA) and Y2O3 contents on the sintering properties of Al2O3–AlN system were carefully compared, in terms of X-ray diffraction (XRD), microstructure, density, X-ray photoelectron spectroscopy (XPS) and optical transmittance. According to the XPS and XRD lattice analysis, the chemical structure of the materials was not obviously affected by different dopants”.
  • Line 23–26. “Obvious enhancement was achieved using Li+ aids from the result that Li: AlONs showing higher transparency and less optical defects. A higher LA content of 20 wt% was more effective to removing pores and obtaining higher transmittance (~86.8% at ~3.5 μm). Thus, pores were the main contribution for the difference between the samples with different aids”.
  • Line 15–16. “The purpose of this work was to explain the difference between the sintering aids of Li+ and Y3+ on AlON ceramic fabrication via reaction sintering method”.

Comment 3: the lengthy sentences may be split in to smaller sentence without change of its meaning.

Response:

Some lengthy sentences have been split into shorter sentences with same meaning as shown next.

  • Line 58–60. “Study [16] indicated that Y2O3 below 0.6 wt% could extremely promote the sintering reaction. However, excess Y2O3 would react with Al2O3 to form Y3Al5O12 (YAG) phase, which could limit the grain growth of AlON ceramics”.
  • Line 189–192. “For the observation, the subsequent decreases of densities from 1550 °C to 1600 °C in Figure 1a could be attributed to the formation of AlON phases. Meanwhile, volume expansion occurred due to the solid-state reaction between AlN and Al2O3”.
  • Line 229–231. “In combination with the results (Figure 2–3), it was concluded that, LA could prevent the sintering at lower temperature (below 1650 °C). However, the sintering process was obviously facilitated at higher temperature [26]”.

Comment 4: Also, suggested to include the recent references in the introduction part.

Response:

Two new literatures have been added in the introduction part.

Line 65: Ref. 17 and ref. 18 (Zhou et al, Ceram. Int. 2022, 48, 24788–24792; Chen et al, J. Adv. Ceram. 2022, 11, 1153–1162) have been added.

Comment 5: the results and discussions part should be compared with the literature data. To redo the part of results and discussions by a systematic presentation of the results by which the readers of the articles manage to follow the article more easily.

Response:

The comparisons of the present results of XRD and XPS with other publications have been added.

  • Line 206– 207:The words “XRD patterns of single AlON phases of four samples exhibited good similarity with previous reports about Y: AlON [14, 16] and Li:AlON [27]” are added in the text.
  • Line 257– 266:The words “Further insights into the chemical structure of the ceramics were investigated by XRD lattice measurement and the XPS spectra. The lattice parameters of four obtained samples in Figure 4a–d have been calculated, as shown in Table 2. The values were 7.93–7.94 Å, without obvious difference between the four samples. As shown in Figure 5a, there was also no obvious difference between the XPS patterns. The bonding energy of Al 2p were 74.32, 74.28, 74.28 and 74.38 eV (Figure 5b), respectively. These bonding energy values were also very close to the literature reports [39, 40]. It seemed that the chemical structure of the materials was not obvious changed by these dopants.” are added in the text.
  • Three sections are divided, as follows: 3.1. Sintering and densification; 3.2. Phase and microstructural evolution; 3.3. Optical properties and material structure.

Comment 6: compare XRD results with other articles.

Response:

The revision was made as reply to comment 5.

Comment 7: in discussions about structural analysis, XPS measurements are missing. Let these measurements be added to have a substrate for the correlation of the discussions.

Response:

  • The first revision was made as reply to comment 5.
  • The experimental results of XPS (Figure 5) have been added in section 3.2.
  • The corresponding references ([14, 16, 27], [39, 40]) have also been renewed in Reference.

Comment 8: Calculate the XRD parameters (crystallite size, network parameter, volume, etc.) and enter in a table. It can be discussed how the network parameters increase for the majority fax which becomes unique at higher temperatures

Response:

  • The XRD parameters (network parameter and volume) of four obtained samples in Figure 4a-d have been calculated. The results have been added in Table 2.
  • Line 259–262: A description of Table 2 has also been added in revised manuscript. “The XRD parameters of four obtained samples in Figure 4a–d have been calculated, as shown in Table 2. The lattice parameters and volume were about 7.93–7.94 Å and 499–500 Å3, respectively. The network parameters showed no obvious difference between the compositions”.
  • The crystallite size was hard to obtain by XRD result. As well known, the method to obtain the crystallite size was not so precise when grain size was larger than 100 nm.

Comment 9: SEM figure quality is poor throughout. To improve the quality of the figures. Enlarge the characters in the figures.

Response:

The characters of all SEM figures have been enlarged in the revised manuscript. The figure quality has been enhanced.

Comment 10: Conclusions should be short with important observations.

Response: Conclusions have been shortened.

  • The words “Y: AlON and Li: AlON were fabricated via reaction sintering method using Y2O3–AlN–Al2O3 and LA–AlN–Al2O3 raw powders. The influences of the Y2O3 and LA dopants on the microstructure and optical properties could be summarized as follows” have been delected.
  • Line 308–320: Related revisions are presented.

Comment 11: References are not written according to the guide (not all authors are listed in all references, they are not marked with initials, the titles of the articles and magazines are not mentioned in many places, the volumes or pages are missing).

Response:

Authors, titles of articles and magazines and volumes of all the references have been carefully checked or corrected especially for [1, 2, 5, 11, 14, 16, 22, 26].

Reviewer 2 Report

This work is devoted to the study of the influence of various sintering aids on the optical properties of AlON ceramics. The work has undoubted relevance, the narrative is logical and consistent. The conclusions are supported by the necessary evidence. The work can certainly be published in a journal after some minor corrections, as given below.

1. Introduction

            Page 1, Lines 34-35. „After the fabrication of the first trans-ucent AlON ceramic in 1979 [8].“ – Check this sentence. It doesn't seem to be finished.

            Page 2, Line 66. “…using pressure-less sintering method.” Above in the text you wrote word “pressureless” together.

            Page 2, Lines 66-70. “The reported strength and hardness of Mg: AlON are slightly lower than those of AlON due to ~8% MgO doping.”, „Previously, Li: AlON with higher strength and hardness had been produced by Clay et al. [22] by reaction sintering using LA (also called as zeta alumina), Al2O3 and AlN.“ –  The authors should give the values of the mechanical properties of ceramics for comparison.

            Page 2, Line 69. “…by reaction sintering using LA (also called as zeta alumina)” – a more detailed explanation of LA should be given.

            Page 2, Line 76. “The samples also showed excellent mechanical properties.” – The values of the mechanical properties of the obtained materials should be given.

2. Experiments

            Page 3. Lines 107-108. “Then the samples were sintered at 1600–1800 °C (with the heating rate of 10 °C/min) and kept for 2 h under a N2 atmosphere.” – The samples were sintered in a high-temperature furnace?

            LA is LiAl5O8?

3. Results and discussion

            As reported in the "Materials and Methods" section, sintering was carried out at temperatures of 1600-1800 °C, however, the authors of the article give density indicators for samples obtained at temperatures below sintering temperatures, including at 400 °C. The authors should explain this. Were there any preliminary experiments to determine the density of undersintered samples?

            Page 4. Authors should provide theoretical densities for all compositions.

            Page 6. Section 3.2. The authors found a change in the density of the samples after HIP? If yes, authors should provide these values.

            Figure 3 and Figure 5 provide fracture surfaces of the obtained samples?

            Figure 5. Can the authors once again conclude why, at 20% addition of LA, the sample does not have microdefects, which leads to the achievement of maximum transmittance?

            Figure 6b. What is the reason for the slight drop in the transmittance in the wavelength region of 3.25 μm?

            Page 8. Can the authors once again conclude what advantages they have achieved in comparison with other works?

Author Response

Responses to reviewer #2

Comment 1: Page 1, Lines 34-35. After the fabrication of the first transucent AlON ceramic in 1979 [8].“ – Check this sentence. It doesn't seem to be finished.

Response:

This sentence has been corrected in the revised manuscript.

Line 40-42: “After the fabrication of the first translucent AlON ceramic in 1979 [8], two methods have been developed to prepare AlON ceramics with high transparency over the decades”.

Comment 2: Page 2, Line 66. “…using pressure-less sintering method.” Above in the text you wrote word “pressureless” together.

Response:

Line 73: The word has been corrected as “pressureless” in the revised manuscript.

Comment 3: Page 2, Lines 66-70. “The reported strength and hardness of Mg: AlON are slightly lower than those of AlON due to ~8% MgO doping.”, „Previously, Li: AlON with higher strength and hardness had been produced by Clay et al. [22] by reaction sintering using LA (also called as zeta alumina), Al2O3 and AlN.“ –  The authors should give the values of the mechanical properties of ceramics for comparison.

Response:

Related revisions have been made in revised manuscript as shown next:

  • Line 73-75: “ The reported strength (~280 MPa) and hardness (~13.5 GPa) of Mg: AlON are slightly lower than those of AlON (~310 MPa, 15 GPa) due to ~8% MgO doping [22,23].”
  • Line 75-77: “ Previously, Li: AlON with higher hardness (~15-17 GPa) had been produced by Clay et al. [24] by reaction sintering using LiAl5O8 (LA, also called as zeta alumina), Al2O3 and AlN”

Comment 4: Page 2, Line 69. “…by reaction sintering using LA (also called as zeta alumina)” – a more detailed explanation of LA should be given.

Response:

  • Line 77: The word “LA” has been changed to “LiAl5O8” and its abbreviation “LA” in following bracket.
  • Line 163-165: More detailed explanation has been added. “The role of LA could be accounted by the fact that it could transform into a face-centered cubic structure at above 1290 °C with a similar lattice parameter (a=7.920 Å) to that of AlON” in section 3.1.

Comment 5: Page 2, Line 76. “The samples also showed excellent mechanical properties.” – The values of the mechanical properties of the obtained materials should be given.

Response:

Line 83-84: “The samples also showed higher flexural strength (~ 310 MPa)

Comment 6: Page 3. Lines 107-108. “Then the samples were sintered at 1600–1800 °C (with the heating rate of 10 °C/min) and kept for 2 h under a N2 atmosphere.” – The samples were sintered in a high-temperature furnace? LA is LiAl5O8?

Response:

  • Line 121-125: Related words have been revised as “The obtained green bodies were then calcined at 650 °C for 20 h in air in a muffle furnace to eliminate organic residues. Then the samples were sintered at 400–1800 °C (with the heating rate of 10 °C/min) and kept for 2 h under a N2 atmosphere in a graphite heating furnace. The cooling rates were about 10 °C/min (free cooling below 600 °C).”
  • Line 77: LA is the abbreviation of LiAl5O8. Related revision has been made.

Comment 7: As reported in the "Materials and Methods" section, sintering was carried out at temperatures of 1600-1800 °C, however, the authors of the article give density indicators for samples obtained at temperatures below sintering temperatures, including at 400 °C. The authors should explain this. Were there any preliminary experiments to determine the density of undersintered samples?

Response:

  • Sorry, there was a spelling error in the manuscript. And we have revised it. “Then the samples were sintered at 400–1800 °C (with the heating rate of 10 °C/min) and kept for 2 h under a N2 atmosphere in a graphite heating furnace”.
  • The density of the sintered samples (at different temperatures) was determined when they are cooled to room temperature. The density of the under-sintered samples was hard to measure.
  • Line 124-125: related words were added as “The cooling rates were about 10 °C/min (free cooling below 600 °C).”
  • Line 176: Related words are changed as “Figure 1. The densities a) and photos b) of the samples that sintered under different temperatures”.

Comment 8: Page 4. Authors should provide theoretical densities for all compositions.

Response:

The theoretical densities for all compositions have been added in Table 1 in page 3.

Comment 9: Page 6. Section 3.2. The authors found a change in the density of the samples after HIP? If yes, authors should provide these values.

Response:

  • Yes, there was a slight change of the density after HIP.
  • Line 286-289. The words “What’s more, the densities of these sintered samples after HIP showed a slight increase from 3.69, 3.69, 3.68 and 3.68 g/cm3 to 3.70, 3.70, 3.70 and 3.69 g/cm3, respectively. Lower porosity was hence supported by higher related density (~100%) of LA doped systems.” have been added.

Comment 10: Figure 3 and Figure 5 provide fracture surfaces of the obtained samples?

Response:

  • Yes, they are the fracture surfaces.
  • Related expressions are revised in line 234 and 278.

Comment 11: Figure 5. Can the authors once again conclude why, at 20% addition of LA, the sample does not have microdefects, which leads to the achievement of maximum transmittance?

Response:

  • Firstly, a continuous densification was observed by 20% addition of LA, which was beneficial for densification or elimination of residual pores.
  • Secondly, solid-state reaction between AlN and Al2O3 was accomplished by an intrinsic volume expansion during sintering. In the present research, Y3+ or 10%LA doping was not enough to depress the volume expansion. While 20% addition was enough.
  • Thirdly, LA exhibited a cubic structure and has a similar lattice parameter (7.920 Å) to that of AlON, which was beneficial for solid-solution reaction and formation of AlON. The reaction between AlN and Al2O3 was completed at lower temperature (~1700 ℃) comparing with other compositions.

Comment 12: Figure 6b. What is the reason for the slight drop in the transmittance in the wavelength region of 3.25 μm?

Response:

The reason for the slight absorption in the wavelength region near 3.25 μm has been added in revised manuscript. The words “In addition, the absorption near 3.25 μm was considered to be caused by -OH formed by the adsorption of oxygen and water molecules in air on sample surface [45]” have been added in Line 297-299.

Comment 13: Page 8. Can the authors once again conclude what advantages they have achieved in comparison with other works?

Response:

The fabrication of Y3+ doped AlON with high transparency is still a challenge by reaction sintering method. While for Li+ doped AlON, the ceramics can be transparent. It is therefore of a primary interest to identify the difference by which the microstructure and transparency are influenced by aids in this type of materials. The present work investigated the widely used aids (Y3+ and Li+) to realize the difference between them for AlON based ceramic fabrication. Related studies were not carefully reported previously. It was suggested that, volume expansion and large pores were observed in these Y3+ dopped AlON ceramics, which was responsible for the low transparency of the ceramics. Our work indicated that, the importance of sintering aids should be carefully designed for the fabrication of highly transparent ceramics by solid-state reaction sintering.

Reviewer 3 Report

[1]         Abstract: is good

[2]         Keywords: is good

[3]         Introduction: add the previous work in literature? the objective of the present work carefully.

[4]         Experiments: need revision

[5]         Results and Discussion:

In lines 119-120 FTIR from (2500 to 7000) nm where is the figure?

Figure 4 optical properties its very useful If the author add the other optical parameters such as n, k, er, ei, Eg … etc

[6]         Figures: Figure 1: revise 1550 oC for all temperatures, Figures 2, 3, 4, 5 and 6 are good

[7]         Tables: Table 1 is better to write (5) beside the component for example AIN(%), Al2O3(5) … and so on

[8]         Conclusion: is good

[9]         References: cite the following recent references

DOI: https://doi.org/10.1016/j.jallcom.2017.09.084

DOI: https://doi.org/10.1016/j.jallcom.2017.02.117

Best Regards

Author Response

Responses to reviewer #3

Comment 1: Introduction: add the previous work in literature? the objective of the present work carefully.

Response:

  • Line 84-87: The words are revised as “In our previous work [28], Li: AlON with higher transmittance (~86.8%) and flexural strength (~332 MPa) was fabricated by reaction sintering of LA–Al2O3–AlN composites. The work compared the sintering of LA doped and aid-free AlON ceramics”.
  • Lines 389-391. Ref. [28] are added.
  • Line 15-16: The words “The purpose of this work was to explain the difference between the sintering aids of Li+ and Y3+ on AlON ceramic fabrication via reaction sintering method” have been added.

Comment 2: Experiments: need revision

Response:

Related revision has been made as shown next.

  • Line 102-106: “The mixed powders were consisted of AlN (10 wt%), Al2O3(varying composition range of 70–90 wt%), Y2O3 (0.5 wt% and 1 wt%, respectively) and LiAl5O8 powders (10 wt% and 20 wt%, respectively). The average particle sizes of AlN, Al2O3, Y2O3 and LiAl5O8 raw powder were about 1 μm, 150 nm, 100 nm and 200 nm, respectively
  • Line 112-119: “The powders were mixed by planetary-ball milling method using absolute ethanol as dispersion medium. One vertical planetary-ball milling grinder with a constant ratio of 2:1 of rotating rated of the jar and sun disk was used in this study. To control the degree of contamination, high quality alumina balls (3.7g/cm3, 99.99%, Φ10 mm, Jingdezhen Betterwear New Materials Co., Ltd., Zhejiang. China) and nylon jar were used. The ratio of ball to powder was 8:1. The milling time and rate were 24 h and 200 rpm, respectively. After milling, the slurry was dried in vacuum at 50 °C for 2 h in a rotary evaporator to evaporate the ethanol, followed by sieved with a 100-mesh sieve.”
  • Line 121-125: “The obtained green bodies were then calcined at 650 °C for 20 h in air in a muffle furnace to eliminate organic residues. Then the samples were sintered at 400–1800 °C (with the heating rate of 10 °C/min) and kept for 2 h under a N2 atmosphere in a graphite heating furnace. The cooling rates were 10 °C/min (free cooling below 600 °C).
  • Line 134-136: “The chemical states of obtained samples with different doping were identified by X-ray photoelectron spectroscopy (XPS, Axis Ultra, Kratos, England).”

Comment 3: Results and Discussion: In lines 119-120 FTIR from (2500 to 7000) nm where is the figure?

Response:

Line 139: The expression has been revised. “…and Fourier transform infrared spectroscopy from 2500 nm to 6500 nm (FT–IR, Model Nexus, Thermo Nicolet Corporation, Madison, WI, USA)”. The corresponding figures are shown in Figs. 4e and 7b.

Comment 4: Figure 4 optical properties its very useful If the author add the other optical parameters such as n, k, er, ei, Eg … etc

Response:

  • We are so sorry that, precise measurement of n, k, er, ei,… towards a bulk material is hard to accomplish, due to the obvious scattering in ceramics especially for samples with lower transparency (Figure 4a-c). The scattering will disturb the measurement results. Even for the sample presented in Figure 7a, the transmittance was not so high enough in the visible band. In the present work, we mainly focus on the fabrication, and the effects of sintering aids on the optical transmittance.
  • The Eg value is near 5.2 eV for all samples according to the optical spectra of Figure 4e and the equation Eg=hcmin. The words “Additionally, the shorter cutoff wavelength was near 0.23 μm for all samples, therefore the bandgaps (Eg) of the materials were ~5.2 eV according to the equation Eg=hc/λmin” have been added in line 266-268.

For the situations, we anticipate to get your understanding and support.

Comment 5: Figure 1: revise 1550 oC for all temperatures, Figures 2, 3, 4, 5 and 6 are good.

Response:

Related revisions have been made in Figure 1.

Comment 6: Table 1 is better to write (5) beside the component for example AIN(%), Al2O3(5) … and so on

Response:

Related revisions have been made in Table 1 and Table 2, according to your kind advice.

Comment 7: cite the following recent references. DOI: https://doi.org/10.1016/j.jallcom.2017.09.084; DOI: https://doi.org/10.1016/j.jallcom.2017.02.117

Response:

  • Line Related references have been added.
  • Line 299-304: The words “It could be anticipated the application of highly transparent AlON based ceramics by scale up the size. For this consideration, precise measurement of optical constants (n, k, Eg, etc.) was also very important for optical application. Related methods have been successfully conducted in thin film materials [46,47]. The method could be a good guidance for us to determine the optical constants of transparent LiAlON in our next work” have been added.

Reviewer 4 Report

The manuscript entitled: “Effects of sintering aids on the microstructure and optical properties of reactively sintered AlON based transparent ceramics” deals with Y: AlON and Li: AlON fabricated by reaction sintering method using Y2O3–AlN–Al2O3 and LA–AlN–Al2O3 raw powders. The influence of the Y2O3 and LA dopants on the microstructure and optical properties were thoroughly examined. The presented study is very interesting and can attract scientific audience. However, there are some changes authors should make before manuscript can be accepted in the Materials journal.

1. The English presentation of the whole manuscript is at very poor level (i.e. lines 34-35, 44-45, 50-51, etc.). The English should be checked by a native English speaker.

2. Section 2.1. Which mill was used for ball milling? Add the specification in the Experimental section.

3. Figure 2. Symbols for phases are too large. Diffractograms cannot be clearly seen. Please, correct it.

4. Lines 199-200. Authors wrote: “No obvious difference was observed between the 10 wt% LA and 20 wt% LA dopped samples at 1500–1650 °C”. However, I could see the difference in the micrographs between LA doped samples at 1600 °C. The sample is obviously less porous.

5. Doped not dopped, correct it throughout the whole manuscript.

6. Line 220. I think it’s Figure 4c-d (not 4d-e).

Author Response

Responses to reviewer #4

Comment 1: The English presentation of the whole manuscript is at very poor level (i.e. lines 34-35, 44-45, 50-51, etc.). The English should be checked by a native English speaker.

Response:

Thanks for your kind advice. The English of the whole manuscript has been carefully checked. Some of the revisions are shown as follows (the location was changed due to major revision).

Line 35-38: “It showed processing flexibility in fabricating large size and complex shape, and exhibited wide range of optical transmittance. Therefore, AlON transparent ceramics were capable for ideal infrared windows, domes and transparent armors [5,6]

Line 48: “Consequently, they could enrich or precipitate on grain boundaries and reduce transmittance obviously”.

Line 53-55: “Y2O3 and MgO co-doped AlON with high transmittance (~86.1%) was also successfully fabricated by Jiang et al. [15] via pressureless sintering (1800 °C, 6h) followed by hot isostatic pressing (HIP) at 1825 °C for 3h”.

Comment 2: Section 2.1. Which mill was used for ball milling? Add the specification in the Experimental section.

Response:

The detailed information of milling in our work has been added in revised manuscript as shown in section 2.1. “The powders were mixed by planetary-ball milling method using absolute ethanol as dispersion medium. One vertical planetary-ball milling grinder with a constant ratio of 2:1 of rotating rated of the jar and sun disk was used in this study. To control the degree of contamination, high quality alumina balls (3.7g/cm3, 99.99%, Φ10 mm, Jingdezhen Betterwear New Materials Co., Ltd., Zhejiang. China) and nylon jar were used. The ratio of ball to powder was 8:1. The milling time and rate were 24 h and 200 rpm, respectively.” (Line 112-117)

Comment 3: Figure 2. Symbols for phases are too large. Diffractograms cannot be clearly seen. Please, correct it.

Response:

The symbols have been corrected as required.

Comment 4: Lines 199-200. Authors wrote: “No obvious difference was observed between the 10 wt% LA and 20 wt% LA dopped samples at 1500–1650 °C”. However, I could see the difference in the micrographs between LA doped samples at 1600 °C. The sample is obviously less porous.

Response:

Thanks very much for your careful attention! For the 20 wt% LA system, the figure of 1600 °C and 1650 °C are reversed due to our careless attention. Furthermore, we really find the slight difference between them at 1650 °C. Now we have revised it.

  • Line 220-223: “No obvious difference was observed between the 10 wt% LA and 20 wt% LA doped samples at 1500–1600 °C. In contrast, a slight difference was presented at 1650 °C. The sample of 20 wt% LA doped sample was less porous”.
  • Correspondingly, Figure 3 was renewed.

Comment 5: Doped not dopped, correct it throughout the whole manuscript.

Response:

Thanks, we have carefully checked spelling error in the whole manuscript and revised it.

Comment 6: Line 220. I think it’s Figure 4c-d (not 4d-e).

Response:

We greatly appreciate your correction about this mistake in the revised manuscript (Line 243).

Round 2

Reviewer 1 Report

The work has been improved, additions have been made, but there are still some missing aspects:

1. To expand the XPS interpretation part by using specialized literature

2. The analyzes are not correlated. There should be a correlation of information between XPS, XRD, TEM and SEM

3. The part of applications is rather poorly discussed without any analysis of the obtained ceramic products. For example CIELAB, BET, porosity measurements.....

4. The degree of novelty is not presented in this review either.

5. To present at the end of the introduction the objectives that led to this study.

6. In the following, the guidelines for technical editing of references are not respected: a comma is inserted after each author (46,47), a period is not placed after each word of the abbreviated journal and a period is not placed before the year, as you have done for all references (Guide: Author 1, A.B.; Author 2, C.D. Title of the article. Abbreviated Journal Name Year, Volume, page range), if there is only one code instead of a page range, use that code and not in parentheses (1 ,2).

7. To improve the English language.

Author Response

Dear reviewer,

Thanks again for your continuous attention on our paper entitled “Effects of Y2O3 and LiAl5O8 on the microstructure and optical properties of reactively sintered AlON based transparent ceramics (materials-1989217)”.

We checked the manuscript according to your professional comments. All the changes or revisions have been marked (highlighted) by red colour in the revised manuscript. We sincerely hope this manuscript will be finally acceptable to publish on “Materials”.

Reviewer’s comments and our corresponding revisions are shown in the appendix.

Please do not hesitate to contact us if you have any suggestion on this paper. We will consider it carefully and reply you as soon as possible.

Yours sincerely,

Yuezhong Wang

------------------------------------------------------------------------------------------------------

Authors:

Guojian Yang, Peng Sun, Yuezhong Wang, Zitao Shi, Qingwei Yan, Shasha Li, Guoyong Yang, Ke Yang, Shijie Dun, Peng Shang, Lifen Deng, He Li and Nan Jiang

Appendix

Responses to reviewer #1

Comment 1: To expand the XPS interpretation part by using specialized literature

Response:

  • Thanks for your mention, a more specialized literature has been introduced to replace the previous [41].
  • Some words are added. Line 264-266: “According to the deep investigation [41], the Al 2p and N 1s XPS showed that the AlON film was composed of Al−N, Al−O, and N−Al−O bonds”.

Comment 2: The analyzes are not correlated. There should be a correlation of information between XPS, XRD, TEM and SEM

Response:

The words are added in this revised manuscript.

Line 294-297: “In combination with the above XRD, SEM and XPS analysis, we could obtain the conclusion that: a) the dopants effected only the microstructural defect (pores) and thus optical transmission; b) the dopants did not change the phase, chemical binding and lattice parameters of the obtained ceramics”.

Comment 3: The part of applications is rather poorly discussed without any analysis of the obtained ceramic products. For example CIELAB, BET, porosity measurements.....

Response:

Really, we have realized that the part of discussion on applications is rather poorly. In the present work, we mainly focused on the fabrication arts of AlON based transparent ceramics by comparing the effects of dopants on the microstructure and optical properties. If we really want to discuss the applications, many data are needed, which could be hard to accomplish due to limited time and budget. So, we deleted related words of “application” after careful consideration. Therefore, revision is made as follows:

Line 304-309: “To end up, we fabricated AlON based ceramic with different transparency, and related mechanism on the effects of dopants were systematically investigated. Li:AlON with high transparency was obtained. Precise measurement of optical constants (n, k, Eg, etc.) could be anticipated in our next work for more insights into this material. Related methods have been successfully conducted in thin film materials [47,48]”.

Comment 4: The degree of novelty is not presented in this review either.

Response:

The novelty has been summarized, as shown in Abstract.

Line 20-22: “We firstly reported that, there was obvious volume expansion in the Y3+ dopped AlON ceramics, which was responsible for the low transparency of the ceramics”.

Comment 5: To present at the end of the introduction the objectives that led to this study.

Response:

Related revision has been made.

Line 91-94: “Presently, it is of a primary goal to identify the difference by which the microstructure and transparency are influenced in this type of materials. Therefore, a comprehensive comparison between Y2O3 and LA dopants for AlON ceramic fabrication by reaction sintering was carried out”.

Comment 6: In the following, the guidelines for technical editing of references are not respected: a comma is inserted after each author (46,47), a period is not placed after each word of the abbreviated journal and a period is not placed before the year, as you have done for all references (Guide: Author 1, A.B.; Author 2, C.D. Title of the article. Abbreviated Journal Name Year, Volume, page range), if there is only one code instead of a page range, use that code and not in parentheses (1 ,2)

Response:

Related revisions have been made according to your kind advice.

Comment 7: To improve the English language.

Response:

The English language has been checked in the whole manuscript.

  • Line 16: “The purpose of this work was to explain the difference between the sintering aids of Li+ and Y3+ on AlON ceramic via reaction sintering method”.
  • Line 17: “The effects of LiAl5O8 (LA) and Y2O3 on the sintering of Al2O3–AlN system…”
  • Line 22-23: “Obvious enhancements were achieved using Li+ aids from the results that Li: AlONs showing higher transparency and less optical defects”.
  • Line 23-24: “A higher LA content (20 wt%) was effective to remove pores and thus obtain higher transmittance (~86.8% at ~3.5 μm)”.
  • Line 24-25: “Thus, pores were the main contributions to the property difference between the dopant samples”.
  • Line 25-27: “The importance of sintering aids should be carefully realized for the reaction sintering fabrication of AlON based ceramics towards high transparency”.
  • Line 33-34: “It was firstly synthesized by Yamaguchi and Yanajida in 1959 [3]”.
  • Line 34: “AlON was a solid-solution with the composition that centered at…”.
  • Line 45: “in reducing pores and promoting densification”.
  • Line 56: “However, the using of Y2O3 or La2O3 dopants was not so…”.
  • Line 60: “Therefore, major obstacles for the elimination of optical defects such as pores and secondary phases…”
  • Line 63: “…sintering method with Y3+ dopant”.
  • Line 64: “…to further reduce the pores”.
  • Line 65: “To obtain high transparency and decrease the sintering temperature”.
  • Line 66: “…such as MgO and LiAl5O8 (LA)”.
  • Line 67: “…AlON could be stabilized using MgO at lower temperature…”.
  • Line 68: “…but the visible band was not high enough (~65%)”.
  • Line 88: “the obtaining of Y3+ doped AlON with high transparency…”.
  • Line 89: “Consequently, a further study on the widely used aids (Y3+ and Li+) is very essential for AlON ceramic fabrication”.
  • Line 96: “…were carefully studied”.
  • Line 99-100: “…were prepared by reaction sintering under nitrogen atmosphere”.
  • Line 143: “a significant increase of the densities was seen…”.
  • Line 144: “For the Y2O3-doped systems”
  • Line 148-149: “The decreasing of densities at high temperature was indicative of volume expansion, since no obvious mass changes were observed in the samples”.
  • Line 150: “For the LA-doped compositions C and D”.
  • Line 154-155: “low content of LA”.
  • Line 169: “…promote the elimination of”.
  • Line 170: “under high temperature [32,33]”.
  • Line 180: “It indicated that”.
  • Line 181: “the shrinkage of the raw powders”.
  • Line 182: “the Y-doped”.
  • Line 183: “LA-doped systems”.
  • Line 186: “detecting limits of the equipment”.
  • Line 193-194: “As the temperature increased”.
  • Line 194: “and AlONs continuously formed”.
  • Line 197-198: “AlON phases firstly formed and then reacted with residual Al2O3. The situation was different from previous reports”.
  • Line 199: “For the Y-doped and 10 wt% LA-doped systems”.
  • Line 200-201: “For 20 wt% LA-doped system”.
  • Line 203: “reports on Y: AlON”.
  • Line 204: “could effectively promote the”.
  • Line 208: “For the Y2O3 doped samples”.
  • Line 210: “high densities”.
  • Line 212: “for the Y2O3 doped systems”.
  • Line 212-214: “However, the promotion effects were prohibited due to microstructural large pores formation at higher temperature (1700 °C)”.
  • Line 214: “the Y2O3 doped and LA doped”.
  • Line 215: “the grain sizes of samples C and D were significantly smaller than”.
  • Line 220: “was promoted by”.
  • Line 223: “10 wt% LA doped sample, the 20 wt% LA doped ceramic”.
  • Line 224: “For reactively sintered systems”.
  • Line 225: “very susceptible to temperature and dopants”.
  • Line 236-237: “showed the appearances of obtained samples”.
  • Line 244: “With 10% LA addition”.
  • Line 245: “For the Y2O3 doped”.
  • Line 247: “Y2O3 doped systems”.
  • Line 248: “the additive-free sample [18,28]”.
  • Line 251: “It indicated that the depressed transmittances were”.
  • Line 283: “Further study from microstructure supported the conclusion that”.
  • Line 285-286: “a plenty of pores were located at the grain boundaries in samples A and B. The residual pores could…”.
  • Line 298: “According to the above results”.

Thanks Again!

Reviewer 2 Report

The authors have significantly improved the manuscript and answered all the questions. I ask the authors to check the English language in detail and add the following information to the final version of the manuscript:

1. Connect the results of X-ray phase analysis with the results of SEM and XPS.

2. Change the caption of Table 2. For example, “Crystal lattice parameters of the AlON samples with different dopants”.

3. Add the reference parameters of the AlON crystal lattice in Table 2 to compare changes in its structure after the introduction of dopants.

Author Response

Dear reviewer,

Thanks again for your continuous attention on our paper entitled “Effects of Y2O3 and LiAl5O8 on the microstructure and optical properties of reactively sintered AlON based transparent ceramics (materials-1989217)”.

We checked the manuscript according to your professional comments. All the changes or revisions have been marked (highlighted) by red colour in the revised manuscript. We sincerely hope this manuscript will be finally acceptable to publish on “Materials”.

Reviewer’s comments and our corresponding revisions are shown in the appendix.

Please do not hesitate to contact us if you have any suggestion on this paper. We will consider it carefully and reply you as soon as possible.

Yours sincerely,

Yuezhong Wang

------------------------------------------------------------------------------------------------------

Authors:

Guojian Yang, Peng Sun, Yuezhong Wang, Zitao Shi, Qingwei Yan, Shasha Li, Guoyong Yang, Ke Yang, Shijie Dun, Peng Shang, Lifen Deng, He Li and Nan Jiang

Appendix

Responses to reviewer #2

Comment 1: Connect the results of X-ray phase analysis with the results of SEM and XPS.

Response:

The words are added in this revised manuscript.

Line 294-297: “In combination with the above XRD, SEM, XPS analysis, we could obtain the conclusion that: a) the dopants effected only the microstructural defect (pores) and thus optical transmission; b) the dopants did not obviously change the phase, chemical binding and lattice parameters of the obtained ceramics”.

Comment 2: Change the caption of Table 2. For example, “Crystal lattice parameters of the AlON samples with different dopants”.

Response:

We have revised the caption as “Crystal lattice parameters of the AlON based ceramic samples with different dopants” (Line 275).

Comment 3: Add the reference parameters of the AlON crystal lattice in Table 2 to compare changes in its structure after the introduction of dopants

Response:

The reference parameters of the AlON crystal lattices [39,40] have been added in Table 2. A comparison has been made after the introduction of dopants. Revisions are made as follows:

  • Line 259-262: “According to previous publications, the parameters of AlONs were mainly dependent on N contents [39,40]. The lower AlN content, the lower lattice parameter and volume. Therefore, a slight difference between the parameters was perhaps due to minor composition difference
  • References: after [40] are renewed.

Thanks Again!
